# Pre-Transplant Alpha-Fetoprotein > 25.5 and Its Dynamic on Waitlist Are Predictors of HCC Recurrence after Liver Transplantation for Patients Meeting Milan Criteria

**DOI:** 10.3390/cancers13235976

**Published:** 2021-11-27

**Authors:** Bianca Magro, Domenico Pinelli, Massimo De Giorgio, Maria Grazia Lucà, Arianna Ghirardi, Alessandra Carrobio, Giuseppe Baronio, Luca Del Prete, Franck Nounamo, Andrea Gianatti, Michele Colledan, Stefano Fagiuoli

**Affiliations:** 1Gastroenterology, Hepatology and Liver Transplantation, Department of Medicine-Papa Giovanni, XXIII Hospital, 24122 Bergamo, Italy; mdegiorgio@asst-pg23.it (M.D.G.); mluca@asst-pg23.it (M.G.L.); sfagiuoli@asst-pg23.it (S.F.); 2Unit of Hepato-Biliary Surgery and Liver Transplantation, ASST Papa Giovanni XXIII, 24122 Bergamo, Italy; dpinelli@asst-pg23.it (D.P.); gbaronio@asst-pg23.it (G.B.); luca.delprete@unimi.it (L.D.P.); fnaounamo@asst-pg23.it (F.N.); mcolledan@asst-pg23.it (M.C.); 3FROM Research Foundation, Papa Giovanni XXIII Hospital, 24122 Bergamo, Italy; aghirardi@fondazionefrom.it (A.G.); acarrobio@fondazionefrom.it (A.C.); 4Pathology Unit, ASST Papa Giovanni XXIII, 24122 Bergamo, Italy; agianatti@asst-pg23.it

**Keywords:** HCC, alpha fetoprotein, liver transplantation, cirrhosis, recurrence

## Abstract

**Simple Summary:**

In the setting of liver transplantation, prediction of hepatocellular carcinoma (HCC) recurrence persists as a fundamental issue, and few tools are available during the waitlist. Biological features such as alpha-fetoprotein values are strong predictors, but it is necessary to understand how we can use them. In this study, an AFP cut-off value was individuated and also its dynamic increase on the waitlist was defined as a predictor.

**Abstract:**

Background and Aim: Hepatocellular carcinoma (HCC) recurrence rates after liver transplantation (LT) range between 8 and 20%. Alpha-fetoprotein (AFP) levels at transplant can predict HCC recurrence, however a defined cut-off value is needed to better stratify patients. The aim of this study was to evaluate the rate of HCC recurrence at our centre and to identify predictors, focusing on AFP. Methods: We retrospectively analysed 236 consecutive patients that were waitlisted for HCC who all met the Milan criteria from January 2001 to December 2017 at our liver transplant centre. A total of twenty-nine patients dropped out while they were waitlisted, and 207 patients were included in the final analysis. All survival analyses included the competing-risk model. Results: The mean age was 56.8 ± 6.8 years. A total of 14% were female (*n* = 29/207). The median MELD (model for end-stage liver disease) at LT was 12 (9–16). The median time on the waitlist was 92 (41–170) days. The HCC recurrence rate was 16.4% (*n* = 34/208). The mean time to recurrence was 3.3 ± 2.8 years. The median AFP levels at transplant were higher in patients with HCC recurrence (*p* < 0.001). At multivariate analysis, the AFP value at transplant that was greater than 25.5 ng/mL (AUC 0.69) was a strong predictor of HCC recurrence after LT [sHR 3.3 (1.6–6.81); *p* = 0.001]. The HCC cumulative incidence function (CIF) of recurrence at 10 years from LT was significantly higher in patients with AFP > 25.5 ng/mL [34.3% vs. 11.5% (*p* = 0.001)]. Moreover, an increase in AFP > 20.8%, was significantly associated with HCC recurrence (*p* = 0.034). Conclusions: In conclusion, in our retrospective study, the AFP level at transplant > 25.5 ng/mL and its increase greater than 20.8% on the waitlist were strong predictors of HCC recurrence after LT in a cohort of patients that were waitlisted within the Milan criteria. However further studies are needed to validate these data.

## 1. Introduction

Hepatocellular carcinoma (HCC) is the fifth most common cancer in men (7.5%) and ninth in women (3.4%). Moreover, it is the second most common cause of death from cancer worldwide, being responsible for nearly 9.1% of the total deaths for cancer in 2012 [1]. It is one of the very few cancers that can be treated by liver transplantation, which represents the only valid treatment for both malignancy and underlying cirrhosis.

The close association between HCC and cirrhosis determines some peculiar characteristics of this neoplasm such as the very high percentage of neoplastic recurrence even after potentially radical treatments. This is due to an intrahepatic metastasis of the primary neoplasm or to a de novo tumorigenesis that occurs in other damaged hepatocytes and the limited access to therapeutic strategies such as surgical resection. In fact, patients with chronic liver disease often have a reduced functional reserve which limits their tolerance even to small volume depletions and also the hepatic regenerative response appears reduced in these patients [2].

Patients that are undergoing LT because of HCC within the Milan Criteria [3], show a four-year survival of 75%, with recurrence rates that are below 15%. These excellent results completely changed the treatment strategy for this malignancy; LT has since become increasingly popular as a treatment for HCC. However, the limited access to the waiting list, together with the progressively more significant drop-out from the list has induced the search for an expansion of the Milan criteria. Indeed, in recent decades, there has been a surge of interest in this field. Several authors have proposed new criteria that are based on anatomical findings that are less restrictive than the Milan criteria, like the University of California San Francisco (UCSF) criteria [4] or in Up-to-7 criteria [5], whereas other authors have introduced histology findings. Moreover, researchers have proposed new criteria including serological markers like α-fetoprotein (AFP) [6] and Des-γ-carboxyprothombin (DCP) [7,8].

Today, the proposed expansion of the criteria is more than ever supported by the advent of new drugs for the curative therapy of hepatitis C.

The treatment of the infection that has been the most significant cause of liver disease in the Western world in the last 30 years, should, in fact, reduce the demand for organs, increasing their availability for other indications [9].

Even though LT plays a central role in the treatment of HCC, recurrence still occurs in 6–18% of patients [10]. Nearly 40–50% of recurrences occur within a year from transplantation and 20% develop during the second year [11]. Several authors [10,12] speculated that an early recurrence (i.e., within one or two years) could be caused by tumor cell engraftment in the new liver by unrecognized extra-hepatic HCC localization, whereas chronic injury due to graft rejection, ischemic cholangiopathy, or a relapse of the primary liver disease could be responsible for late recurrences. According to this hypothesis, the latter would represent de-novo HCCs which a better prognosis [12].

There has been a growing number of publications focusing on factors that are involved in HCC recurrence after LT. Some authors have argued that donors that are older than 60 years [13], cold ischemia time (CIT) more than 10 h, post-reperfusion injury [14], and high graft steatosis [15] could be involved in the recurrence rate. Others stated that HCC not responding to down-staging treatments implies a more aggressive biological behaviour which is associated with higher recurrence post-LT. Vascular invasion and poorly differentiated tumour cells are considered risk factors for recurrence, and the routine evaluation of these data can be useful in the pre-LT decision-making processes [16,17].

Another study added the radiological response to neoadjuvant therapies to improve the prognostic score, Metroticket 2.0 [18]. The study showed that patients with partial response or disease progression should be carefully considered to maintain an acceptably low risk of post-LT HCC-related death.

Moreover, the importance of the pre-LT radiological assessment was also enhanced by an experience of a single centre where the inclusion of every nodule from intermediate to high probability of harboring HCC improved the Metroticket 2.0 accuracy [19].

Little is known about the role of specific immunosuppressive protocols on HCC recurrence after LT. Some studies [20,21] reported an association between the high level of calcineurin inhibitors and early HCC recurrence, whereas, according to others, mTOR inhibitors could exert a protective role [22].

Understanding the complexity of HCC biology is vitally important for establishing effective predictors of recurrence and the identification of these predictors is crucial, moreover if you decide to broaden the transplantation criteria, due to the risk of making the transplant itself futile.

In recent years, the link between biological markers and HCC staging and prognosis has been progressively highlighted. However, a gold standard is still far away from the practical use. Moreover, the only routinely utilized biological parameter (serum AFP) has been removed from most of the guidelines as a diagnostic test for HCC [23,24] because of well documented limitations: not all tumors secrete AFP; serum concentrations can be within the normal range in up to 40% of small HCC; and AFP levels do not correlate with clinical features of HCC, such as tumor size or vascular invasion. Although serum levels of AFP are typically higher in advanced HCC compared to early HCC, higher levels tend to be more specific but less sensitive [25,26].

Despite these limitations, AFP could play an important role in the specific setting of post-LT HCC recurrence. Indeed, an elevated level of AFP prior to LT can predict both the mortality and the recurrence rate of HCC in patients that are undergoing liver transplantation.

The aim of this study was to evaluate the possible predictors of HCC recurrence and HCC-recurrence-free survival in patients undergoing LT within the Milan criteria at our institution.

## 2. Materials and Methods

### 2.1. Study Population

We retrospectively collected data from 236 consecutive patients that were waitlisted for HCC within the Milan Criteria, at our liver transplant centre, from January 2001 to December 2017. A total of 207 patients were included in our analysis. Of those, 21 patients dropped out from the waitlist for disease progression and 8 were removed because an HCC diagnosis was not confirmed. Appendix A. All of the patients that were included in this study met the Milan criteria and presented AFP < 400 ng/mL at transplant. HCC restaging in the waitlist occurred every 3 months, or before if a clinical change was registered.

### 2.2. Data Collection

Pre-LT data included features of the recipients, such as demographic data, clinical and liver -related history, AFP levels, type, and the timing of bridge therapy that was performed for HCC while waitlisted. The donor age, graft quality (included graft steatosis grading), and total ischemia time were also considered, as well as TNM [23], TTV, and the size and number of HCC nodules at pre-surgery imaging. TTV was calculated as the sum of the total tumor nodule volume. HCC was diagnosed according to the Barcelona Clinic Liver Cancer [27] and LIRADS criteria [28]. Child-Pugh class and the model of end-stage liver disease (MELD) score were used to classify the hepatic function of each patient that was enrolled. A liver biopsy was performed in eight cases where imaging was not conclusive between the HCC and cholangiocarcinoma or benign nodule. At the explant, the number and maximum diameter of active HCCs, necrosis of nodules, tumor grading according to modified Edmondson-Stainer criteria [29], and the presence/absence of microvascular invasion were recorded.

The follow-up of these patients always included serial serum AFP measurement, as well as abdominal and chest CT that were performed every three months up to the third postoperative month, and then every 6 months up to the third postoperative year, then at least every year. The date of death, date of HCC recurrence, original disease recurrence or last censoring, and the cause of death were also recorded.

### 2.3. Outcomes

All survival analyses included the competing-risk model. For recurrence-free survival, death without evidence of HCC recurrence was considered as a competing event. The primary outcome was to individuate the reliable predictors of HCC recurrence after LT. For multivariate analyses, all statistically significant variables following univariate analysis (time of waitlist, cirrhosis etiology, and AFP at transplant) were included. We also analysed the dynamic role of AFP on the waitlist as the difference between the AFP value at the moment of the inscription on the waitlist and the last one before liver transplantation. The secondary outcomes were: recurrence-free survival (RFS) that was defined as the survival rate from the time of LT to HCC recurrence, and overall survival (OS) that was defined as the survival from the time of the HCC diagnosis to death (HCC-related and not).

### 2.4. Statistical Analysis

Descriptive statistics were used to summarize the characteristics of patients with or without recurrence. The continuous variables were expressed as a mean and standard deviation (SD) or as a median and interquartile range (IQR), depending on their normal or non-normal distribution. The categorical variables were expressed as absolute counts and percentages.

Bivariate associations of demographic and clinical factors with recurrence were tested using the chi-square test (or Fisher’s exact test when appropriated) for categorical variables and *t*-test (or Wilcoxon-Mann-Whitney test for non-normal distributed variables) for continuous variables.

A multivariable Cox proportional hazard model was used to estimate the predictors of recurrence. The results were presented as a hazard ratio (HR) and 95% confidence intervals (CI). To adjust the estimates for the presence of death as a competing event, the Fine-Gray competing risk model was also implemented. The results were presented as sub distribution hazard ratio (sHR) and 95% CI.

We used the cumulative incidence function (CIF) to plot the cumulative incidence for recurrence, considering death as competing event, and to represent HCC-specific mortality and other causes of death.

Finally, a subgroup analysis was fitted among patients with increased AFP values at transplant to investigate a potential association of ∆AFP with recurrence. ∆AFP was defined as the relative increase in AFP values from entry (E) to transplant (T) [(AFPT-AFPE)/AFPE] * 100. The best cut-off of ∆AFP was identified by the ROC curve (Youden index). For all of the tested hypotheses, two-tailed *p*-values < 0.05 were considered significant.

Statistical analysis was performed using Stata Software, release 16 (StataCorp LP, College Station, TX, USA).

## 3. Results

### 3.1. Patients Recipients

Baseline characteristics are shown in Table 1.

The mean age was 56.8 ± 6.8 years. A total of 29 patients were female (14%; *n* = 29/207). Of the patients, 35 had a Child-Pugh class C (16.9%) at LT. Median MELD at LT was 12 (9–16). A total of 14 patients had MELD > 25. The median time on the waitlist was 92 (41–170) days: 67 patients (32.4%) 60 days; 54 patients (26.1%) between 61 and 120 days; 49 patients (23.7%) between 121 and 240 days and 37 patients (17.9%) > 241 days.

The median time of follow up was 5.53 years (IQR: 2.93–9.64).

The median AFP value at transplant was 9.7 ng/mL (4.1–27.1 ng/mL). The most common cause of liver disease was HCV infection (*n* = 106/207; 51.2%), followed by HBV infection (*n* = 62/207; 30.4%).

Imaging nearby the LT showed in 135 patients (65.2%) a single nodule of HCC, 2 nodules in 51 patients (24.6%), and 3 nodules in 21 patients (10.1%). 

The median diameter of the nodule value among the larger nodules was 2.4 cm (1.8–3 cm), the median value of diameter addition was 3 cm (2–4 cm). The TTV median value was 8.4 cm^3^ (4.2–15.9 cm^3^).

A total of 86% of the patients (*n* = 178/208) underwent bridge therapy while they were waitlisted: 9 hepatic resections (4.3%), 9 ethanol injection (4.3%); 130 percutaneous radiofrequency ablations (RFA) (62.8%); 91 trans arterial chemoembolization (TACE) (44%).

Likewise, a total of 86% (*n* = 178/208) of the recipients were on calcineurin-inhibitors (CNI) as backbone immunosuppressive treatment after LT.

### 3.2. Donors

The baseline characteristics are shown in Table 1.

The mean age was 58.7 ± 17.6 years. In 180 patients (87%), steatosis was inferior to 30%. The mean ischemic total time was 394.8 ± 116.3 min.

A total of five transplant procedures utilized a split liver (7.2%).

### 3.3. Histological Features of Explanted Livers

In 167 cases (80.7%), the total number of nodules at explant was 3; in 40 patients (19.3%) was > 3. The median value of the major nodule diameter was 2.5 cm (2–3.5 cm). The vascular micro-invasion was detected in 29 patients (14%). Total necrosis was found in 38 patients (18.4%), partial necrosis in 129 patients (62.3%) and no tumor necrosis was found in 19 livers (9.2%). Table 1.

### 3.4. Primary Outcome

The predictors of HCC recurrence after LT.

HCC recurrence rate was 16.4% (*n* = 34/207). The mean time to recurrence was 39.3 ± 33.6 months. HCC hepatic recurrence was recorded in 11/34 patients (32.3%), five in the first two years.

In 17/34 (50%) patients, HCC recurred only extrahepatically. In 6 patients (17.6%), the time to recurrence was within the first year, in 8 (23.5%) within the second year, and in 20 (58.8%) in the third year.

The results of the multivariate analysis are shown in Table 2. We included in the analysis the significative variables at univariate analysis (Table 1).

At multivariate analysis, the AFP value at transplant was > 25.5 ng/mL (AUC 0.69), Appendix A, was a strong predictor of HCC recurrence after LT [sHR 2.5 (1.6–6.81); *p* = 0.01]. The time of waitlist of <6 months was not confirmed in the multivariate analysis as a predictor. 

Previous HCC bridge therapies were not predictors of HCC recurrence in our cohort.

### 3.5. Patients with Increased AFP on Waitlist

We analysed patients with an AFP increase while waitlisted (*n* = 75). A cut-off of 20.8% increase was identified by the corresponding ROC curve (AUC = 0.69; 95% CI 0.55–0.83) in Appendix A. A subgroup analysis was fitted, and among patients with an increase in AFP > 20.8% at transplant, HCC recurrence after liver transplantation was significantly higher (*p* = 0.034). Table 3.

### 3.6. Secondary Outcomes

HCC recurrence free survival (RFS) was stratified for AFP values.

HCC RFS was 55.6% (95% CI 47.3–63.1%) after 10 years from LT.

We stratified the analysis according to the AFP value and HCC CIF of recurrence at 10 years from LT was significant higher in patients with an AFP > 25.5 [34.3% (95% CI 21.0–48.0%) vs. 11.5% (6.3–18.4%) (*p* = 0.001)]. Figure 1.

### 3.7. Overall Survival

The overall survival at 10 years from LT was 59% (50–67%). We also analysed the cause-specific mortality: HCC-related mortality was 14%, while all other causes of mortality was 27% at 10 years from LT; Figure 2, Appendix A.

## 4. Discussion

LT is widely validated as a treatment for HCC, but HCC recurrence is an important negative predictor of post-transplant survival, affecting around 10–20% of the cases [1,30].

In this setting, it is paramount to identify both patients who can benefit most, as well as avoiding “organ-futility”. Several studies have focused their interest on finding the ideal model to predict post-LT HCC recurrence.

Initially, the Milan criteria in 1996 included only the tumor burden and number of nodules at explant [3]. In 2012, a multi-centre French study incorporated the AFP threshold, number of nodules, and the largest tumor diameter in a prognostic score. In particular, among patients within the Milano Criteria where a score greater than 2 identified patients with AFP levels greater than 1000 ng/mL that were at high risk of recurrence (*p* = 0.006) [31].

Our study shows that RFS at 10 years after LT in patients within the Milano Criteria was 55%, as previously demonstrated in the literature [3].

Following univariate analysis, the waitlist length (*p* = 0.009), AFP value at transplant (*p* = 0.001), and the cirrhosis etiology (*p* = 0.008) emerged as possible predictors of HCC recurrence; HCV etiology was not a predictor. Previous studies have demonstrated the association between several other factors with post-LT recurrence.

The RETREAT [32] score showed elevated AFP, the presence of MVI on the explant, and the largest viable tumor diameter plus the number of viable tumors on the explant as possible prognostic factors. Microvascular invasion (MVI) on the explant can predict an approximately 3.8 to 4.9-fold increase in HCC recurrence [29,30,33]. We could not confirm these findings in our study group, because they failed to reach the statistical significance (*p* = 0.06).

The recent XXL trial [34] in a group of waitlisted patients beyond the Milan criteria showed that effective down staging treatment correlates significantly with a higher tumor event-free survival after LT (*p* = 0.003), while, as demonstrated in previous studies, the radiological response to downstaging could improve the prognostic score [18].

In the setting of patients that are affected by HCC on the waitlist for LT, the concept of prioritization is often debated. Some authors have suggested a new paradigm in which the response to bridging or downstaging represent the main drivers for patient selection and priority allocation. Both the response or the lack of response to neoadjuvant therapies could positively impact prioritization, always within the eligibility criteria of LT [34].

This approach was also studied by Di Sandro et al., where the authors showed how that were patients at higher risk on the waitlist could beneficiate from an early LT (within two months), while for patients with an intermediate risk it would be better to wait to evaluate tumor behavior [35].

In our study, the execution of bridge therapy to transplant with loco regional treatment did not impact recurrence rates (*p* = 0.34). It is worth noting that one of the strengths of the present single-centre study is represented by the presence of a multidisciplinary team, with stable and coherent selection criteria: it is somewhat expected that bridge procedures on Milan-in HCC waitlisted patients might not significantly influence HCC post-LT recurrence.

The use of mammalian target of rapamycin (mTOR) inhibitors as the main immunosuppressive agents are associated with a lower incidence of HCC recurrence after LT. The SILVER study showed an improvement in recurrence-free survival in the first three to five years in patients that were treated with sirolimus vs calcineurin inhibitors (CNI) [36].

In our study, following univariate analysis, patients that were in treatment with mTOR had a higher rate of HCC recurrence (*p* = 0.03): However, we must consider that, in most cases, the switch of immunosuppression to mTOR was made after the diagnosis of post-LT HCC recurrence.

Recently, several findings suggest the resurgent role of AFP as a powerful prognostic factor for HCC recurrence post-LT [37,38]. In this setting, it is the only validated predictor that we can use before LT, as MVI and tumor grading are evident only at the moment of the explant.

However, there is no consensus on the applicable cut-off value for improving a patient’s selection and planning a proper HCC recurrence surveillance after LT as yet. AFP values between 16 and 320 ng/mL seem to be associated with poor post-LT outcomes [37].

Moreover, it has been shown that extremely elevated AFP levels (>1000 ng/mL) before LT is associated with the worst post-LT survivals, regardless of the tumor burden [39]. However, it rests an undefined area in which we cannot know what happens in terms of recurrence and survival and it would be better to stratify HCC patients before LT. In a recent retrospective study of 3819 patients who underwent LT, in downstaging groups, AFP < 100 ng/mL was the only independent predictor of HCC recurrence (HR 2.6; *p* = 0.02) [40].

In our multivariate analysis only, the AFP values at transplant were confirmed as HCC recurrence predictors: indeed, AFP > 25.5 ng/mL at the time of LT predicted a greater than three-time increased risk of post-LT HCC recurrence (sHR 3.3; *p* = 0.01). Moreover, the subgroup analysis showed that an increase in AFP values > 20.8% on the waitlist correlated with higher HCC recurrence rates (*p* = 0.03). These findings confirm the pivotal role of AFP as an “alarm bell” of HCC recurrence in Milan-in-HCC LT recipients.

This result was also recently demonstrated in a large study that was conducted on 2236 patients that were transplanted for HCC. The authors validated the NYCA score that included the dynamic AFP response (AFP-R). This study highlighted the utility of the dynamic AFP-R at predicting the patient outcomes after liver transplant for HCC [41].

Regarding the role of AFP in this setting, Mahmud et al. showed that, in a cohort of 1164 transplanted patients, pre-transplant AFP < 500 ng/mL had a 1.6-fold higher risk of death versus those with AFP < 20 ng/mL (*p* < 0.001) [37].

Moreover, Koch et al. showed that in patients with an AFP ratio (AFP at recurrence divided by AFP 3 months before recurrence) of 0.5, HCC recurrence was greater than 70 months, compared to a median of 8 months in patients with a ratio of 0.5. It underlines that a rise in AFP levels could identify patients that are at short-term risk for HCC recurrence after LT [38].

The AFP threshold that was identified in this study could optimize both patient selection and also improve surveillance protocols after LT. Post-LT surveillance should be intensified during the period of the highest post-LT recurrences; unfortunately very few studies were aimed at this issue, which leaves the frequency and timing of both imaging and serum testing unclear.

At present, there is no specific evidence-based risk stratification criteria that can be strongly recommended. In general CT or MRI should be performed every six months for 2–3 years after LT, and AFP every six months for 5 years post LT [42].

Our results confirm the potential role of AFP to reduce HCC recurrence and improve the outcomes after LT; it is fundamental to understand the importance of AFP to avoid transplant “futility” and to make decisions in a multidisciplinary setting.

This study presents several limitations. Firstly was the selection of patients that were represented only by patients within the Milan Criteria. This reflects the past policy of the transplant centre; in fact, 21 patients were excluded from the analysis for disease progression. Secondly, the relatively small sample size and also only the two measurements of AFP during the waitlist as well as the retrospective design. On the other hand, a single-centre study should usually warrant homogeneity. We believe that our results could contribute to stratify LT recipients and eventually improve post-LT outcomes.

## 5. Conclusions

In conclusion, in our retrospective study, an AFP value at transplant >25.5 ng/mL and its dynamic increase that was greater than 20.8% while on the waitlist, were predictors for HCC recurrence after LT in a cohort of patients within the Milano criteria. The AFP cut-off could help to stratify patients but it is its increase on the waitlist that could highlight that patients needed more attention on two sides. It could be a reason to prioritize patients for LT, maybe representing the “favorable window” to avoid future graft futility for disease progression. On the other hand, if the increase continues to persist and patients are always eligible for LT, physicians have to optimize immunosuppression and strictly monitor the recurrence after LT. These data could be used to correctly stratify HCC patients before LT and improve surveillance after LT, but further studies are needed to validate these results.

## Figures and Tables

**Figure 1 cancers-13-05976-f001:**
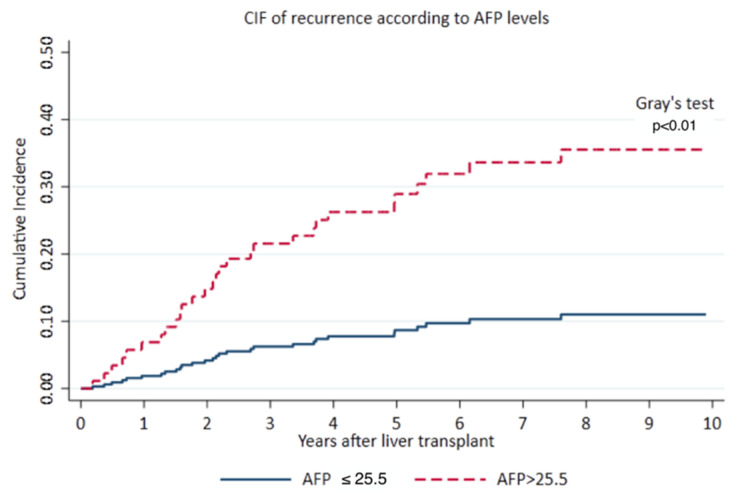
10-year Cumulative incidence function (CIF) of recurrence according to AFP level.

**Figure 2 cancers-13-05976-f002:**
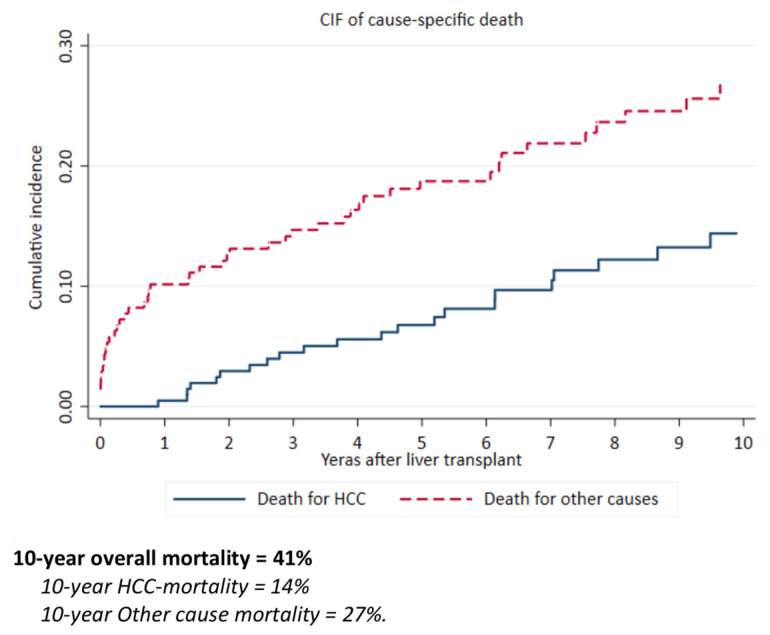
Cumulative Incidence Function (CIF) of cause-specific death.

**Table 1 cancers-13-05976-t001:** Baseline characteristics of recipients and donors according to HCC recurrence.

	Recurrence		*p* *
	No (*n* = 173)	Yes (*n* = 34)	
Donor’s characteristics			
Age, media ± SD	58.3 ± 17.8	60.7 ± 16.9	0.454
Steatosis			
>50	4 (2.3%)	0 (0.0%)	0.785
30–50	4 (2.3%)	1 (2.9%)	
<30	152 (87.9%)	28 (82.4%)	
NA	13	5	
Ischemia time, mean ± SD	392.5 ± 110.9	406.7 ± 141.9	0.516
Recipient’s characteristics			
Age, mean ± SD	56.9 ± 6.6	55.9 ± 7.9	0.431
Gender			
Male	145 (83.8%)	33 (97.1%)	0.055
Female	28 (16.2%)	1 (2.9%)	
BMI, median (Q1, Q3) NA	25.4 (23.2, 27.4) 1	27.0 (23.3, 30.1) 0	0.115
Child-Pugh class			
A	69 (39.9%)	15 (44.1%)	0.379
B	67 (38.7%)	16 (47.1%)	
C	32 (18.5%)	3 (8.8%)	
NA	5	0	
Waitlist, months			
≤60	60 (34.7%)	7 (20.6%)	0.049 *
61–120	48 (27.7%)	6 (17.6%)	
121–240	39 (22.5%)	10 (29.4%)	
>241	26 (15.0%)	11 (32.4%)	
Median time of waitlist (Q1, Q3)	92.0 (41.0, 170.0)	172.0 (70.0, 312.0)	0.009 *
MELD score NA	12.0 (9.0, 16.0) 9	12.0 (10.0, 14.0) 0	0.765
Bilirubin NA	1.5 (1.0, 2.8) 37	2.0 (1.3, 2.5) 4	0.672
INR NA	1.3 (1.2, 1.5) 20	1.3 (1.2, 1.6) 2	0.637
Creatinine NA	0.8 (0.7, 0.9) 23	0.8 (0.7, 0.9) 2	0.804
AFP at transplant NA	8.4 (3.8, 21.3) 15	27.5 (7.6, 87.2) 3	0.001 *
Cirrhosis etiology			
HCV	83 (48.0%)	23 (67.6%)	0.008 *
HBV	60 (34.7%)	3 (8.8%)	
HBV and HCV	6 (3.5%)	2 (5.9%)	
Others	20 (11.6%)	6 (17.6%)	
NA	4	0	
Pre-LT characteristics			
Number of nodules			
1	111 (64.2%)	24 (70.6%)	0.709
2	43 (24.9%)	8 (23.5%)	
3	19 (11.0%)	2 (5.9%)	
Largest tumour diameter(cm)	2.4 (1.8, 3.0)	2.5 (2.0, 3.0)	0.364
Sum of the diameters	3.0 (2.0, 4.0)	3.0 (2.0, 4.0)	0.950
TTV	8.2 (4.2, 15.2)	10.7 (4.2, 18.3)	0.314
Bridge therapies	147 (85.0%)	31 (91.2%)	0.34
Number, mean ± SD	1.2 ± 0.8	1.4 ± 1.0	0.15
Surgery	8 (4.6%)	1 (2.9%)	0.66
Alcoholization	6 (3.5%)	3 (8.8%)	0.16
RFA	106 (61.3%)	24 (70.6%)	0.30
TACE	74 (42.8%)	17 (50.0%)	0.44
Post LT characteristics			
Number of nodules			
1–3	143 (82.7%)	24 (70.6%)	0.103
>3	30 (17.3%)	10 (29.4%)	
Largest tumour diameter (cm) NA	2.5 (2.0, 3.5) 1	2.9 (2.0, 3.7) 0	0.237
Microvascular invasion			
No	148 (85.5%)	26 (76.5%)	0.214
Yes	22 (12.7%)	7 (20.6%)	
NA	3	1	
Necrosis			
Total	36 (20.8%)	2 (5.9%)	0.069
Partial	103 (59.5%)	26 (76.5%)	
Absent	17 (9.8%)	2 (5.9%)	
NA	17	4	
Immunosuppression			
CNI	145 (83.8%)	33 (97.1%)	0.57
NA	20	0	
mTOR	25 (16.2%)	11 (32.4%)	0.031 *
NA	19	0	
MMF	23 (14.9%)	3 (8.8%)	0.35
NA	19	0	
Median time from transplant to latest therapy, (months) Q1–Q2	12.3 (6.8–22.0)	10.8 (6.5–19.1)	0.65
Median time from latest therapy to last fup, (years) Q1–Q2	6.3 (4.2–9.8)	6.4 (4.5–8.8)	0.82

AFP: alpha-fetoprotein; CNI: calcineurin inhibitor; HBV: hepatitis B virus; HCV: hepatitis C virus; INR: international normalized ratio; MELD: model of end-stage liver disease; MMF: mycophenolate mofetil; mTOR: mammalian target of rapamycin inhibitors; Q1: first quartile; Q3: third quartile; RFA: radiofrequency ablation; SD: standard deviation; TACE: trans-arterial chemoembolization; TTV: total tumor volume. * Chi-squared test for categorical variables (or Fisher’s exact test, if needed); Student’s *t*-test or Mann-Whitney U test for continuous variables.

**Table 2 cancers-13-05976-t002:** Multivariable Fine–Gray proportional sub-distribution hazard model for recurrence after liver transplant (death was considered as a competing event).

	sHR (95% CI)	*p*
AFP at transplant		
≤25.5	1.00 (reference)	
>25.5	2.50 (1.20–5.21)	0.014
MVI	2.13 (0.90–5.04)	0.085
Etiology HBV (vs. others)	0.33 (0.10–1.13)	0.077
Waitlist		
≤6 months	1.00 (reference)	
>6 months	1.76 (0.85–3.64)	0.127

sHR: Subdistribution Hazard Ratio; CI: Confidence Interval.

**Table 3 cancers-13-05976-t003:** The association between ∆AFP and recurrence in patients with increased AFP values at transplant.

	Total	No Recurrence	Recurrence	*p*-Value
	*n* = 75	*n* = 62	*n* = 13	
∆AFP ≥ 20.8%	43 (57.3%)	32 (51.6%)	11 (84.6%)	0.034

∆AFP was defined as the relative increase in AFP values from entry (E) to transplant (T) [(AFP_T_ − AFP_E_)/AFP_E_] × 100. Cut-off of 20.8% was identified by the corresponding ROC curve (AUC = 0.69, 95% CI 0.55–0.83).

## Data Availability

The data presented in this study are available on request from the corresponding author.

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
