# Peer review of "Pre-Transplant Alpha-Fetoprotein > 25.5 and Its Dynamic on Waitlist Are Predictors of HCC Recurrence after Liver Transplantation for Patients Meeting Milan Criteria"

_cancers, 2021, doi:10.3390/cancers13235976_

Round 1
Reviewer 1 Report
To the authors:
“Pre-transplant alpha-fetoprotein>25.5 and its dynamic on waitlist are predictors of HCC recurrence after liver transplantation for patients meeting Milano criteria”
My comments are as follows.
- Please revise “Child-Pugh score” to “Child-Pugh class” in Table 1.
- Please revise “2.4. STATISTICAL ANALYSIS” to “2.4. Statistical analysis” in “2. Materials and Methods”.
- How did you judge whether alcoholization be present or not?
- Please describe “Time of waitlist” in Table 1 with months, like Table 2.
- Please display the graphs of the ROC curve for setting the cutoff value of AFP at trans-plant to 25.5 ng/mL and the cutoff value of ΔAFP ≥ 20.8%.
- Please show the detailed breakdown of all causes-mortality.
I hope that my comments will be useful in improving the article.
Author Response
Dear Editors
Please find enclosed our Manuscript cancers-1464737 entitled "Pre-transplant alpha-fetoprotein>25.5 and its dynamic on waitlist are predictors of HCC recurrence after liver transplantation for patients meeting Milano criteria” which we hereby resubmit for revision, after taking into consideration the suggestions of the reviewers.
We were pleased to receive your letter and the constructive comments from your reviewers who have shown their interest in our case, and have clearly helped us to improve the paper. We have addressed all their comments and made all the modifications required in the revised version herewith. Our point-by-point responses to these comments, and the details of these changes, are listed below.
We look forward hearing from you, in the hope that your response will now be positive with respect to publication.
Sincerely yours,
Dr. Bianca Magro
Corresponding Author: Bianca Magro, MD
Gastroenterology, Hepatology and Liver Transplantation
Department of Medicine
Papa Giovanni, XXIII Hospital,
Piazza OMS 1, Bergamo, Italy.
Phone: +39 3925277473. Fax: +39 0916552276
E-mail: bianca_magro@hotmail.it
REVIEWER 1
- Please revise “Child-Pugh score” to “Child-Pugh class” in Table 1.
Thank you for your comment. We correct “Child-Pugh score” to “Child-Pugh class” in table 1 as well in the main text for consistency.
- Please revise “2.4. STATISTICAL ANALYSIS” to “2.4. Statistical analysis” in “2. Materials and Methods”.
Thank you for your comment. We made the correction
- How did you judge whether alcoholization be present or not?
Thank you for your comment. Every recipient underwent a thorough selection for liver transplant, including nutritional laboratory tests and psychological to exclude at best of our knowledge alcohol consumption. I agree with you that it could be tricky, however we try to build a relationship of trust based on several meeting with hepatologist and psychologists.
- Please describe “Time of waitlist” in Table 1 with months, like Table 2.
Thank you for your comment. Corrected.
- Please display the graphs of the ROC curve for setting the cutoff value of AFP at trans-plant to 25.5 ng/mL and the cutoff value of ΔAFP ≥ 20.8%.
Thank you for your comment we added them as supplementary material
Figure S1. Receiving operating characteristics curve of AFP at transplant for the discrimination of cases of recurrences (AUC: 0.69, 95% CI: 0.58-0.80)
Figure S2. Receiving operating characteristics curve of relative increase in AFP values from entry to transplant (∆AFP) for the discrimination of cases of recurrences (AUC=0.69, 95% CI 0.55 - 0.83)
- Please show the detailed breakdown of all causes-mortality.
Thank you for your comment we added it as supplementary material
Table S1. All causes of mortality
Causes |
N |
DEATH TOT (N=75) |
HCC |
22 |
29,3% |
NOT HCC |
53 |
70,7% |
PNF |
2 |
2,7% |
Infections |
11 |
14,7% |
Surgical |
1 |
1,3% |
Cardiovascular |
12 |
16,0% |
Neoplasia |
12 |
16,0% |
Cirrhosis recurrence |
9 |
12,0% |
Others |
6 |
8,0% |
REVIEWER 2
1) Please, correct the title switching "MilanO" into "Milan"
Thank you for your comment. We corrected this in both title and main text for consistency.
2) There are several typos throughout the text that must be fixed (ie. at line 45 "within" is repeated twice; "strategy and treatment"at line 46 should be inverted, ad so on) - I suggest another round of proof-reading before resubmission
Thank you for your comment. The manuscript has been revised by all authors and the typos corrected. Furthermore we rewrote a couple of sentence to make the manuscript easier to be read.
3) I would suggest to add a CONSORT diagram for patient selection
Thank you for your comment we added the diagram in the supplementary material
Figure S1. Patient selection
4) The effect of aFP pre-transplant in risk assessment had no impact on donor pool (line 64-65) - please, correct.
Thank you for your comment and important correction. We deleted that sentence. We agree with the reviewers that aFP pre-transplant in risk assessment had no impact on donor pool.
5) There is some problem with the references: reference 7 deals with HCV therapy and not PIVKA, and reference 8 deals with post-transplant recurrence; please cross check.
Thank you for your comment. We checked all the references and correct the mistake.
6) The pivotal role of radiological assessment of tumor biology after downstaging (10.1016/j.jhep.2020.03.018) as well as the radiological tumor features (10.1111/tri.13983) should at least be mentioned in the introduction and further discussed in the discussion session.
Thank you for the comment. We added these sentences in the section introduction:
Another study added the radiological response to neoadjuvant therapies to improve the prognostic score Metroticket 2.0 (ref 10.1016/j.jhep.2020.03.018). The study showed that patients with partial response or disease progression should be carefully considered to maintain an acceptably low risk of post-LT HCC-related death.
Moreover, the importance of the pre-LT radiological assessment was also enhanced by an experience of a single centre, where the inclusion of every nodule from intermediate to high probability of harboring HCC improved the Metroticket 2.0 accuracy (10.1111/tri.13983).
We added also these sentences in the section discussion: while as demonstrated in previous studies radiological response to doswnstaging could improve the prognostic score.

Reviewer 2 Report
In their paper entitled "Pre-transplant alpha-fetoprotein>25.5 and its dynamic on waitlist are predictors of HCC recurrence after liver transplantation for patients meeting Milano criteria" Magro and coll. reported a single center retrospective analysis of the impact of aFP on the oncological outcomes of liver transplantation for HCC
The problem of post-transplant recurrence is of pivotal interest for clinicians and researchers worldwide, therefore the subject of this research is of particular interest.
There are several issues that must be addressed before publication:
1) Please, correct the title switching "MilanO" into "Milan"
2) There are several typos throughout the text that must be fixed (ie. at line 45 "within" is repeated twice; "strategy and treatment"at line 46 should be inverted, ad so on) - I suggest another round of proof-reading before resubmission
3) I would suggest to add a CONSORT diagram for patient selection
4) The effect of aFP pre-transplant in risk assessment had no impact on donor pool (line 64-65) - please, correct.
5) There is some problem with the references: reference 7 deals with HCV therapy and not PIVKA, and reference 8 deals with post-transplant recurrence; please cross check.
6) The pivotal role of radiological assessment of tumor biology after downstaging (10.1016/j.jhep.2020.03.018) as well as the radiological tumor features (10.1111/tri.13983) should at least be mentioned in the introduction and further discussed in the discussion session.
7) The authors should better emphasise the role of patient prioritisation while on the waiting list, as many of the cited the prognostic models fails to assess the transplant priority (10.1002/hep.28420; 10.3390/cancers11060741).
8) The footnotes of Table 1 are in Italian. Please correct, adding the abbreviation legend.
9) The study excluded all Milan-out HCC patients, even if downstaged
Considering that downstaging is an essential strategy in common clinical practice with excellent results in selected patients, the Authors should discuss this choice (I would better say this limitation) in the introduction and discussion section (as well as provide the number of Milan-out patients that were excluded).
10) Please define TTV (total tumor volume?) in the methods section.
11) Please expand the methods section providing a more detailed description of patient management: how was HCC diagnosed (EASL criteria?, LIRADS?)? how many biopsies? how many of these patients underwent salvage LT after previous curative treatment? how often were restaged while on the waiting list?
10) The Authors should provide the time from the last bridging-downstaging in Table 1.
11) How did the Authors choose the risk factors for recurrence?
12) Please report median follow-up.
13) Last, I must point out that (even if statistically significant) an aFP value of 25.5 is a very low cut-off for patient selection, so I would suggest a more moderate attitude towards this finding.
On the other hand, the evidence of aFP increase as a risk factor for tumor recurrence is really interesting, but how would you handle this data? Would you wait longer for achieving a better test of time? would you perform other bridging? Please speculate on this behalf in the discussion/conclusion.
Congratulations to the Authors for their efforts
Best regards.
Author Response
Dear Editors
Please find enclosed our Manuscript cancers-1464737 entitled "Pre-transplant alpha-fetoprotein>25.5 and its dynamic on waitlist are predictors of HCC recurrence after liver transplantation for patients meeting Milano criteria” which we hereby resubmit for revision, after taking into consideration the suggestions of the reviewers.
We were pleased to receive your letter and the constructive comments from your reviewers who have shown their interest in our case, and have clearly helped us to improve the paper. We have addressed all their comments and made all the modifications required in the revised version herewith. Our point-by-point responses to these comments, and the details of these changes, are listed below.
We look forward hearing from you, in the hope that your response will now be positive with respect to publication.
Sincerely yours,
Dr. Bianca Magro
Corresponding Author: Bianca Magro, MD
Gastroenterology, Hepatology and Liver Transplantation
Department of Medicine
Papa Giovanni, XXIII Hospital,
Piazza OMS 1, Bergamo, Italy.
Phone: +39 3925277473. Fax: +39 0916552276
E-mail: bianca_magro@hotmail.it
REVIEWER 2
1) Please, correct the title switching "MilanO" into "Milan"
Thank you for your comment. We corrected this in both title and main text for consistency.
2) There are several typos throughout the text that must be fixed (ie. at line 45 "within" is repeated twice; "strategy and treatment"at line 46 should be inverted, ad so on) - I suggest another round of proof-reading before resubmission
Thank you for your comment. The manuscript has been revised by all authors and the typos corrected. Furthermore we rewrote a couple of sentence to make the manuscript easier to be read.
3) I would suggest to add a CONSORT diagram for patient selection
Thank you for your comment we added the diagram in the supplementary material
Figure S1. Patient selection
4) The effect of aFP pre-transplant in risk assessment had no impact on donor pool (line 64-65) - please, correct.
Thank you for your comment and important correction. We deleted that sentence. We agree with the reviewers that aFP pre-transplant in risk assessment had no impact on donor pool.
5) There is some problem with the references: reference 7 deals with HCV therapy and not PIVKA, and reference 8 deals with post-transplant recurrence; please cross check.
Thank you for your comment. We checked all the references and correct the mistake.
6) The pivotal role of radiological assessment of tumor biology after downstaging (10.1016/j.jhep.2020.03.018) as well as the radiological tumor features (10.1111/tri.13983) should at least be mentioned in the introduction and further discussed in the discussion session.
Thank you for the comment. We added these sentences in the section introduction:
Another study added the radiological response to neoadjuvant therapies to improve the prognostic score Metroticket 2.0 (ref 10.1016/j.jhep.2020.03.018). The study showed that patients with partial response or disease progression should be carefully considered to maintain an acceptably low risk of post-LT HCC-related death.
Moreover, the importance of the pre-LT radiological assessment was also enhanced by an experience of a single centre, where the inclusion of every nodule from intermediate to high probability of harboring HCC improved the Metroticket 2.0 accuracy (10.1111/tri.13983).
We added also these sentences in the section discussion: while as demonstrated in previous studies radiological response to doswnstaging could improve the prognostic score.
7) The authors should better emphasise the role of patient prioritisation while on the waiting list, as many of the cited the prognostic models fails to assess the transplant priority (10.1002/hep.28420; 10.3390/cancers11060741).
Thank you for the comment. We added these sentences in te discussion section: In the setting of patients affected by HCC on waitlist for LT, the concept of prioritization is often debated. Some authors had suggested a new paradigm in which the response to bridging or downstaging represent the main drivers for patient selection and priority allocation. Both the response or the lack of response to neoadjuvant therapies could positively impact prioritization, always within the eligibility criteria of LT. 10.1002/hep.28420
This approach was also studied by Di Sandro et al, the authors in their study showed how patients at higher risk on waitlist could beneficiate of an early LT (within 2 months), while for patients with an intermediate risk it would be better to wait to evaluate tumor behavior. cancers11060741).
8) The footnotes of Table 1 are in Italian. Please correct, adding the abbreviation legend.
Thank you for your comment. Corrected and add the abbreviations.
9) The study excluded all Milan-out HCC patients, even if downstaged
Considering that downstaging is an essential strategy in common clinical practice with excellent results in selected patients, the Authors should discuss this choice (I would better say this limitation) in the introduction and discussion section (as well as provide the number of Milan-out patients that were excluded).
Thank you for the comment. We added this sentence in the discussion section: This study presents several limitations, firstly patients’ selection that is represented only by patients within Milan Criteria, this reflects the past policy of the transplant centre, in fact 21 patients were excluded from the analysis for disease progression. Secondly the relatively small sample size, then the only two measurements of AFP during the waitlist and finally the retrospective design.
10) Please define TTV (total tumor volume?) in the methods section.
Thank you for your comment. TTV is defined as was calculated as the sum of the nodule volumes considering the radius of each nodules. We added this definition in methods section.
11) Please expand the methods section providing a more detailed description of patient management: how was HCC diagnosed (EASL criteria?, LIRADS?)? how many biopsies? how many of these patients underwent salvage LT after previous curative treatment? how often were restaged while on the waiting list?
Thank you for pointing this out. We actually think paragraph is important and would add value to the manuscript. HCC was diagnosed according with Barcelona Clinic Liver Cancer and LIRADS criteria, according to the era considered in our cohort, whereas United Network Organ Sharing (UNOS) and Tumor Node Metastasis (TNM) criteria were used for the stadiation of the disease. Child-Pugh and MELD score were used to classify hepatic function of each patient enrolled. A liver biopsy was performed everytime we had doubts on the nodule nature, for differential diagnosis with cholangocarcinoma or benign nodule. This resulted in the dropped out from the waiting list of eight patients because of HCC diagnosis was not confirmed. HCC restaging occurred every 3 months or before if a clinical change in the patient was registered.
Among the 207 patients enrolled, 178 (86%) underwent bridge therapy while waitlisted: 9 hepatic resections (4.3%), 9 ethanol injection (4.3 %); 130 percutaneous radiofrequency ablations (RFA) (62.8%); 91 trans arterial chemoembolization (TACE) (44%).
We added two references in methods section (27;28)
12) The Authors should provide the time from the last bridging-downstaging in Table 1.
Thank you for your comment. We modified Table 1
13) How did the Authors choose the risk factors for recurrence?
Thank you for the comment.
We modified the multivariate analysis including the significative variables at univariate analysis, but the results didn’t change. We added this sentence in results discussion : We included in the analysis the significative variables at univariate analysis
sHR (95% CI) |
P |
|
AFP at transplant |
||
<=25.5 |
1.00 (Rif.) |
|
>25.5 |
2.50 (1.20-5.21) |
0.014 |
MVI |
2.13 (0.90-5.04) |
0.085 |
Etiology B (vs. others) |
0.33 (0.10-1.13) |
0.077 |
Waitlist |
||
<=6 months |
1.00 (Rif.) |
|
> 6 months |
1.76 (0.85-3.64) |
0.127 |
14) Please report median follow-up.
Thank you for your comment. We reported median time of follow up in results section
Median time of follow up: 5.53 years (IQR: 2.93- 9.64)
15) Last, I must point out that (even if statistically significant) an aFP value of 25.5 is a very low cut-off for patient selection, so I would suggest a more moderate attitude towards this finding.
On the other hand, the evidence of aFP increase as a risk factor for tumor recurrence is really interesting, but how would you handle this data? Would you wait longer for achieving a better test of time? would you perform other bridging? Please speculate on this behalf in the discussion/conclusion.
Thank you for the interesting comment. We modified the conclusion section:
In conclusion, in our retrospective study AFP value at transplant > 25.5 ng/ml and its dynamic increase greater than 20.8% on waitlist, are predictors for HCC recurrence after LT in a cohort of patients within Milano criteria. AFP cut-off could help to stratify patients, but it is its increase on waitlist that could highlight patients needed more attention on two sides: it could be a reason to prioritise patients for LT, maybe representing the “favorable window” to avoid a future graft futility for disease progression; on the other hand, if the increase continues to persist and patients is always eligible for LT, physicians has to optimize immunosuppression and strictly monitor recurrence after LT.These data could be used to stratify correctly HCC patients before LT and to improve surveillance after LT, but further studies are needed to validate these results.

Round 2
Reviewer 1 Report
Comments to Author: The manuscript has been revised well.
Reviewer 2 Report
The Authors properly addressed my initial concerns, improving the paper quality, that is now suitable for publication
Best regards.